# Longitudinal Survey of Coronavirus Circulation and Diversity in Insectivorous Bat Colonies in Zimbabwe

**DOI:** 10.3390/v14040781

**Published:** 2022-04-09

**Authors:** Vimbiso Chidoti, Hélène De Nys, Valérie Pinarello, Getrude Mashura, Dorothée Missé, Laure Guerrini, Davies Pfukenyi, Julien Cappelle, Ngoni Chiweshe, Ahidjo Ayouba, Gift Matope, Martine Peeters, Elizabeth Gori, Mathieu Bourgarel, Florian Liégeois

**Affiliations:** 1Faculty of Veterinary Science, University of Zimbabwe, Harare P.O. Box MP 167, Zimbabwe; vimbisochidoti@gmail.com (V.C.); valerie.pinarello@cird.fr (V.P.); getrudemashura@gmail.com (G.M.); dmpfukenyi@vet.uz.ac.zw (D.P.); giftmatope@gmail.com (G.M.); gori.elizabeth@gmail.com (E.G.); 2ASTRE, CIRAD, INRAE, University of Montpellier, 34980 Montpellier, France; helene.de_nys@cirad.fr (H.D.N.); laure.guerrini@cirad.fr (L.G.); julien.cappelle@cirad.fr (J.C.); mathieu.bourgarel@cirad.fr (M.B.); 3CIRAD, UMR ASTRE, Harare, Zimbabwe; chiweshengoni@gmail.com; 4MIVEGEC, University of Montpellier, IRD, CNRS, 34394 Montpellier, France; dorothee.misse@ird.fr; 5CIRAD, UMR ASTRE, 34398 Montpellier, France; 6TransVIHMI, University of Montpellier, IRD, Inserm, 34394 Montpellier, France; ahidjo.ayouba@ird.fr (A.A.); martine.peeters@ird.fr (M.P.)

**Keywords:** bat coronavirus (Bt CoVs), human–bat interaction, genetic diversity, reproductive phenology, Zimbabwe

## Abstract

Background: Studies have linked bats to outbreaks of viral diseases in human populations such as SARS-CoV-1 and MERS-CoV and the ongoing SARS-CoV-2 pandemic. Methods: We carried out a longitudinal survey from August 2020 to July 2021 at two sites in Zimbabwe with bat–human interactions: Magweto cave and Chirundu farm. A total of 1732 and 1866 individual bat fecal samples were collected, respectively. Coronaviruses and bat species were amplified using PCR systems. Results: Analysis of the coronavirus sequences revealed a high genetic diversity, and we identified different sub-viral groups in the *Alphacoronavirus* and *Betacoronavirus* genus. The established sub-viral groups fell within the described *Alphacoronavirus* sub-genera: *Decacovirus*, *Duvinacovirus*, *Rhinacovirus*, *Setracovirus* and *Minunacovirus* and for *Betacoronavirus* sub-genera: *Sarbecoviruses*, *Merbecovirus* and *Hibecovirus*. Our results showed an overall proportion for CoV positive PCR tests of 23.7% at Chirundu site and 16.5% and 38.9% at Magweto site for insectivorous bats and *Macronycteris gigas*, respectively. Conclusions: The higher risk of bat coronavirus exposure for humans was found in December to March in relation to higher viral shedding peaks of coronaviruses in the parturition, lactation and weaning months of the bat populations at both sites. We also highlight the need to further document viral infectious risk in human/domestic animal populations surrounding bat habitats in Zimbabwe.

## 1. Introduction

The *Coronaviridae* family is a monophyletic cluster in the order of *Nidovirales*. These are enveloped positive stranded RNA viruses of three classes of vertebrates: mammals (corona- and toroviruses), birds (coronaviruses) and fish (bafiniviruses) [1]. They are classified into four genera: *Alphacoronavirus* and *Betacoronavirus,* hosted by mammals; *Gammacoronavirus* and *Deltacoronavirus*, infecting avian species [2,3].

Coronaviruses are etiological agents of respiratory, enteric, hepatic and neurological diseases in animals and humans [4], with variable severity, from asymptomatic to severely ill individuals [5]. Surveillance of coronaviruses in wild animals has led to the discovery of a high diversity of viruses in bats and avian species, suggesting these animals as the natural reservoirs [6].

Seven coronaviruses have been identified in humans [3]. The first human coronaviruses were discovered in the early 1960s, nCoV-OC43 and HCoV-229E [7,8], and they are known to cause mild respiratory disease [9]. HCoV-NL63 and -HKU1 were discovered in 2004 and 2005, respectively [10], they also associated with mild respiratory disease [9]. Other high profile human disease outbreaks of the respiratory system caused by coronaviruses took place during the last two decades, namely severe acute respiratory syndrome type 1 and 2 (SARS-CoV-1 and SARS-CoV-2) and Middle East respiratory syndrome (MERS-CoV) [11,12,13]. 

Phylogenetic evidence shows that HCoV-NL63 and -229E originated from bat coronaviruses [7,8], whereas HCoV-OC43 and HCoV-HKU1 originated from rodents [8,14]. Bats are major and ancient reservoirs of several viral families [15], and recent molecular studies demonstrated bats as natural hosts of important zoonotic viruses such as *Filoviruses* and *Paramyxoviruses* among others [16].

Insectivorous bats have recently been shown to harbor the precursor of severe acute respiratory syndrome coronavirus (SARS-CoV-1) [17]. Studies have linked bats (*Rhinolophus* spp.) to SARS-CoV-1 outbreak in humans, transmitted through an intermediate host, i.e., civet cats [18]. Recent research on the outbreak of SARS-CoV-2 (COVID-19) suggest that bats are the original source of the coronavirus due to genetic similarity of the virus to other known bat coronaviruses [12,19,20]. Bat coronaviruses have also been identified in numerous insectivorous and frugivorous bats in Asia, Africa, America and Europe [21,22].

Bats are the second most diverse mammalian order after rodents [23], and the only order of mammals capable of flying. Bats constitute the order Chiroptera [23] and are classified into two groups: *Yinpterochiroptera*, megabats and a number of microbats, and *Yangochiroptera*, which mainly covers insectivorous microbats [24,25]. Bats possess characteristics that maximize their effectiveness as reservoir hosts. These include high species diversity, long life span, long flight distance and dispersal, interspecies dense roost aggregations, social behaviors, hibernation and torpor as well as unique innate immunology [26,27]. 

The circulation of viruses in bat populations is affected by seasonality and the prevailing environmental conditions [23,28]. Each of these directly affects reproduction periods, gregarious behavior, torpor, population density and rate of contacts between individuals in a population [27,28] consequently determining reproductive number of viruses and their transmission [28]. Hence, risks of viral spillover to other animal species and humans are strongly influenced by these factors. The effect of seasonality on viral circulation has been shown in several studies on coronaviruses in bats [29,30]. Temporal variations associated with the bat reproduction phenology have been shown in several coronavirus studies across the world in bats, with higher coronavirus prevalence detected in juvenile and sexually immature individuals [31,32].

Previous studies carried out in Zimbabwe included the characterization of *Dicistrovirus, Coronavirus* and *Paramyxovirus* in insectivorous bats (*Hipposideros caffer*) [33,34]. More than 60 bat species have been recorded in Zimbabwe and in rural areas, and people are in close contact with these animals [35], posing a risk of spillover of bat-associated viruses to humans. Temporal variations associated with the bat reproduction phenology have been shown in several coronavirus studies across the world in bats, with higher coronavirus prevalence detected in juvenile and sexually immature individuals [31,32]. The prevalence and genetic diversity of coronaviruses circulating in insectivorous bat colonies and their seasonal pattern are not well established. This study sought to establish the prevalence and the genetic diversity of coronaviruses in insectivorous bats according to bat reproduction phenology in two multi-species bat sites in Zimbabwe.

## 2. Methods

### 2.1. Study Sites

The study was conducted in Hurungwe district of Mashonaland West province in Zimbabwe, at two sites, Magweto cave and Chirundu farm, known for their colonies of insectivorous bats. The two sites situated in communal lands represent habitats with close human–bat interactions (Figure 1).

### 2.2. Magweto Cave

Magweto cave site has been subject to preliminary research on virus circulation in *Hipposideros caffer* bats [33]. The site is considered to be a sacred place by the local community. The cave itself is used for various activities such as cultural and religious ceremonies. People also collect guano (bat feces) and use it as fertilizer in their fields, while others sell it. Additionally, *Macronycteris gigas* bats are also present in this cave and are hunted for consumption.

### 2.3. Chirundu Farm

Chirundu farm site is surrounded by a banana plantation where a large group of people work daily. The building basement in which the bats roost is surrounded by a nursery of young trees. Due to irrigation of the plants, the tunnels are always flooded with water most part of the year. The bats fly around the property and feed on insects around the plantation.

### 2.4. Sample and Data Collection

On both sites, non-invasive sampling of bat feces was performed from August 2020 to July 2021 at monthly intervals. Two-square-meter plastic sheets were placed under the bat roosts overnight (5 sheets per site). Feces were collected in the mid-morning from each plastic sheet. Individual sampling was carried out by assuming that a space of 20 by 20 cm on the plastic sheet was representative of the defecation area of one individual: One dropping of fresh feces was selected and placed in 2 mL tubes with 0.5 mL of home-made RNA stabilization solution (http://www.protocol-online.org/prot/Protocols/RNAlater-3999.html; accessed on 1 May 2020). Samples were then stored at −80 °C before further laboratory analyses. During sampling, other observations were recorded such as females showing pregnancy, puppies attached to mothers and puppies in specific roost nurseries. This information was used to map a timeline of bat reproductive phenology. Periods of gestation, parturition, lactation, weaning and presence of 4–6 month old juveniles were determined based on observations (from captures and observation of the roosting bats) combined with the literature to confirm observations or to complete observation gaps (regarding periods or species) [35].

Additionally, two capture sessions were performed at each site In October 2020 and March 2021 to assess the bat diversity and reproductive status. Bats were captured using nest and hard traps. All animals were released after collection of dried blood spot, fecal and salivary swab samples as well as morphometric measurements [36]. 

### 2.5. RNA/DNA Extraction

Both RNA and DNA were extracted from 200 µL of fecal samples using 5X MagMax Pathogen RNA/DNA Kit (Thermo Fisher Scientific, Illkich, France) by using the fecal samples preserved in 0.5 mL of home-made preservative RNA solution. The feces were vortexed vigorously (30 Hz) using a Retsch MM400 Tissue lyser for 5 min to fully homogenize and mix well the fecal particles, followed by centrifugation at 16,000× *g* for ~3 min to fully separate the supernatant from the fecal pellet. A total of 130 µL of the supernatant was collected to isolate and purify the nucleic acids using a Mag Max extraction kit (Thermo Fisher Scientific, France) with the automatic KingFisher Duo Prime Purification System (Thermo Fisher Scientific, France) following the manufacturer’s instructions. A final volume of 80 µL of eluted RNA/DNA was collected and stored at −80 °C.

### 2.6. Reverse Transcription (RT) Using Random Hexamers

A total of 5 µL of RNA/DNA sample template was reverse transcribed using 1 µL random hexamers, 0.5 µL Oligo dT primer, 0.4 µL of dNTPs (10 mM) (Thermo Fisher Scientific, France) and 5.5 µL molecular grade water and incubated for 5 min at 65 °C. Next, 4 µL of Buffer 5×, 2 µL of 0.1 M DTT (M-MLV Reverse Transcriptase, Thermo Fisher Scientific, France) and 1 µL of RNAse OUT were added to the previous mix. After incubation for 2 min at 37 °C, 1 µL of M-MLV reverse transcriptase (M-MLV Reverse Transcriptase, Invitrogen, Thermo Fisher Scientific, Illkrich, France) was added to the mixture followed by incubation at 25 °C for 10 min, 37 °C for 50 min and 70 °C for 15 min. The cDNA obtained was stored at −20 °C.

### 2.7. RNA-Dependent RNA-Polymerase (RdRp) Nested PCR on Coronaviruses Using Pan-Coronavirus Primers

The CoV partial *RdPd* gene was amplified by nested PCR. Both PCR rounds were realized in 50 µL reaction mixture containing 5 µL of cDNA template (first round) and 5 µL of first round amplified product (second round), 0.5 µL DNA polymerase, 5 µL Buffer 10X, 4 µL MgCl₂ (25 mM) (Firepol DNA Polymerase, Solis Bio Dyne, Tartu, Estonia), 0.4 µL dNTPs (25 mM) (Thermo Fisher Scientific, Illkrich, France), 31.1 µL of molecular grade water and 2 µL of pan-coronavirus primers (10 mM) (Thermo Fisher Scientific, Illkrich, France): Pan-**CoV-F1:** 5′-GGKTGGGAYTAYCCKAARTG-3′ and Pan-**CoV-R1:** 3′TGYTGTSWRCARAAYTCRTG-5′ for the first PCR round, Pan-**CoVF2:** 5′GGTTGGGACTATCCTAAGTGTGA-3′, Pan-**CoVR2:** 3′-CCATCATCAGATAGAATCATCAT-5′ for the second PCR round as previously described [37]. First round PCR conditions were as follows: 95 °C, 2 min denaturation followed by 40 cycles at 95 °C, 20 s denaturation, 50 °C, 30 s hybridization, 72 °C, 2 min elongation and a final extension of 72 °C for 5 min. Second round PCR conditions were as follows: 95 °C, 2 min denaturation, 40 cycles at 95 °C, 30 s denaturation, 52 °C, 30 s hybridization, 72 °C, 1 min elongation and a final extension of 72 °C for 5 min. For all PCRs, positive and negative controls were run in parallel. Visualization of positive PCR product was carried out with gel electrophoresis using ethidium bromide on a 1% gel. PCR positive products (440 bp) were directly purified (GeneClean Turbo Kit, MP Biomedicals, Illkirch, France) according to supplier’s instructions and then sequenced using the Sanger sequencing method in both the 5′ and 3′ directions (LGC, Berlin, Germany).

### 2.8. Genetic Analyses

#### 2.8.1. Viruses Sequence Identification and Phylogenetic Analysis

Generated Sanger viral sequence assemblage was performed using Geneious software package V. 2021.2.2 (Biomatters Ltd., Auckland, New Zealand). The generated contigs were compared to a database of sequences online using Basic Local Alignment Search Tool (BLAST) (https://blast.ncbi.nlm.nih.gov/Blast.cgi) to identify the amplified sequence. Mega 7 [38] was used to align the CoV sequences obtained in this study with references obtained from GenBank. A maximum likelihood phylogenetic tree of aligned CoV sequences was implemented by using IQ-Tree [39]. The reliability of branching orders was tested using the bootstrap approach (1000 replicates), and the GTR + F+ I substitution model was determined as the best suited evolution model [39].

#### 2.8.2. Bat Species Identification

Genotyping was carried out only on *Coronavirus* positive samples for bat species identification. Partial amplification of cytochrome B and 12S RNA mitochondrial genes followed by sequencing was used for identification of bat species or genus, respectively, for all the CoV positive samples from the two sites. Cytochrome B/12S RNA partial sequences were amplified by one step PCR in 50 µL reaction mix containing 5 µL of RNA template, 0.5 µL DreamTaq DNA polymerase, 5 µL DreamTaq Buffer 10× (DreamTaq DNA Polymerase, Thermo Fisher Scientific, US), 0.4 µL dNTPs (25 mM) (Thermo Fisher Scientific), 35.35 µL of molecular grade water and 2 µL of primers (10 mM) (Thermo Fisher Scientific, France): **C****ytb-L14724:** 5′-CGAAGCTTGATATGAAAAACCATCGTTG-3′, **Cytb-H15506:** 5′- AGTGGRTTRGCTGGTGTRTARTTGTC-3′ [40]. For the **12S RNA, L1091:** (5′-AAAAAG-CTTCAAACTGGGATTAGATACCCCACTAT-3′) and **H1478:** (5′-TGACTGCAGAGGGTGACGGGCGGTGTGT-3′) primers were used [41]. The PCR reaction was carried out under the following conditions: 95 °C, 3 min denaturation, 20 cycles at 94 °C, 20 s denaturation, 45 °C, 30 s hybridization, 72 °C, 1 min 30 s, followed by 20 cycles at 94 °C, 20 s denaturation, 50 °C, 30 s hybridization, 72 °C, 1 min 30 s elongation and a final extension of 72 °C for 5 min. Visualization of positive PCR product was performed with gel electrophoresis using ethidium bromide on a 1% gel. PCR positive products for cytochrome B (800 bp) and for 12S RNA (400 bp) were directly purified (GeneClean Turbo Kit, MP Biomedicals, Illkirch, France) according to supplier’s instructions and then sequenced using Sanger sequencing method in both the 5′ and 3′ directions (LGC, Germany). Cytochrome B and 12S RNA sequences obtained were then compared to bat species sequences available in the GenBank database using BLAST tool (https://blast.ncbi.nlm.nih.gov/Blast.cgi). 

#### 2.8.3. Temporal Variations of Coronavirus Prevalence and Bat Reproduction Phenology

The prevalence was calculated at the community level for insectivorous bat species from the two sites. The *M. gigas* prevalence was calculated separately as fecal samples could easily be attributed to this species given their size and location in the cave. The reproductive phases of the bats were allocated according to field observations and also the literature review. The proportion of RNA CoV positive individuals was calculated by month and site with 95% confidence intervals (CIs) using the Wilson score test [42]. Descriptive analysis of prevalence of CoVs per month/per season at each site was performed using R software (R-studio) programming to construct box plots. The boxplots were plotted with the x-axis as the prevalence of CoVs and y-axis as the date and reproductive season when sampling occurred at each site. Similar analysis was also carried out to describe the effect of reproductive seasonality on the detection of the sub-viral groups described. Sub-viral groups with more than 30 sequences were analyzed; only Chirundu site analyses were performed as at Magweto there were insufficient data to allow an accurate conclusion to be drawn.

In addition, the influence of the different phases of the bat reproduction cycle (pregnancy, parturition/lactation, weaning, weaned juveniles of 4 to 6 months old) on the prevalence of coronaviruses at each site (Magweto and Chirundu) was tested by running two generalized linear mixed models (GLMMs). Parturition and lactation were merged because our sampling sessions did not allow the separation of the two periods. The response variable with a binomial distribution was the coronavirus positivity of the samples, and the explanatory variables with a fixed effect were the different phases of the reproduction cycle (coded as 1 if it was during the corresponding reproduction phase and 0 if it was not). To account for clustered samples collected during the same sampling session at the same site, we included the session identification code as a random effect to control for repeated measures from the same trapping session.

#### 2.8.4. GenBank Accession Numbers 

The Coronavirus sequences have been deposited to GenBank under the following numbers: OM469940-OM470469.

The Cytochrome B sequences have been deposited to GenBank under the following numbers: OM487705-OM488020.

## 3. Results

### 3.1. Sampling, Data Collection and Reproductive Phenology

Due to travel restrictions linked to COVID-19 in Zimbabwe, no samples were collected in January, April or June 2021 at Chirundu farm and in Magweto none were collected from December 2020 to February 2021 or in May and July 2021. We analyzed 1866 fecal samples from different insectivorous bat species including 308 fecal samples from Macronycteris gigas species collected at Magweto cave site and a total of 1732 fecal samples from insectivorous bats collected at the Chirundu farm site (Table 1).

At both sites, the reproductive season, i.e., from gestation to weaning, was observed from September to February for the predominant insectivorous bat species. At both sites, the different insectivorous bat families observed were found to be synchronous regarding their reproductive cycles, and consensus reproduction periods were determined based on the literature and observations (Table 1 and Appendix A). The larger insectivorous *M. gigas* is a migratory bat species and was mainly present at Magweto cave from October 2020 to March 2021 during gestation, parturition and 4–6 month juvenile periods, and the reproductive cycle was also synchronous with that of the smaller insectivorous bats (Table 1 and Appendix A).

### 3.2. Bat Species

By using *BLAST* to compare the newly obtained 800 bp *Cytochrome B* sequences of the CoV positive samples to the *CytB* sequences available in GenBank we identified different bat species at the two sampling sites. We identified at least seven species; *Hipposideros caffer* and *Miniopterus mossambicus* were present at both sites. At Chirundu farm, we also identified *Rhinolophus landeri and Nycteris thebaica*, whereas at Magweto cave we also characterized *Macronycteris gigas, Cleotis percivali* and *Rhinolophus simulator*. The presence of these species was confirmed by capture sessions. We also characterized bats from the *Rhinolophus* genus, close to *Rhinolophus eloquens* from the captured animals. Results obtained by BLAST analysis were confirmed by maximum likelihood phylogenetic analysis (Appendix A).

### 3.3. Prevalence and Seasonality of RNA Coronaviruses at the Bat Community Level 

The overall prevalence of coronaviruses at Chirundu farm site was 23.7% (95% CI: 22.08–26.13). The lowest prevalence was 1.35% (95% CI: 0.5–3.4) observed in samples collected in October 2020, and prevalence increased between December and March 2021, corresponding to the lactation, weaning and 4–6 month juvenile periods, with the highest prevalence of 44.19% (95% CI: 37.4–45.2) in February, corresponding to the weaning period (Table 1, Figure 2A). At Magweto cave site the sampling prevalence was calculated for the insectivorous bat species and the *Macronycteris gigas* separately. For *M. gigas* the overall prevalence of CoVs was 38.9% (95% CI: 33.68–44.51) with a peak of 56.4% (95% CI: 47.7–64.9) in October 2020 during the gestation period (Table 1, Figure 2B). For the bat species which comprised *Hipposideros* spp., *Rhinolophus* spp., *Nycteris* spp. and *Miniopterus* spp., the overall prevalence was 16.5% (95% CI:14.74–18.42) with the lowest prevalence (2%; 95% CI: 0.98–4.1)) detected in September 2020 and the highest prevalence detected in November 2020 (35.9%; 95% CI: 30–42.4), corresponding to the period of gestation and parturition, and March 2021 (34.7%; 95% CI: 28.9–40.9), which corresponds to the juvenile periods (Table 1, Figure 2C). Prevalence during the weaning period for this site remains unknown as no samples were collected during that time.

Results from the first GLMM showed higher coronavirus prevalence associated with the parturition/lactation (odds ratio (OR) = 5.57, 95% CI) = (1.16–29.56), *p*-value = 0.012) and weaned juveniles of 4 to 6 month old periods (OR = 4.22 (0.83–21.35, *p* = 0.037)) at the Magweto site. Results from the second GLMM showed higher coronavirus prevalence associated with the parturition/lactation period (OR = 6.04 (1.85–22.62, *p* = 0.0014)) and the weaning period (OR = 5.29 (1.16–26.56, *p* = 0.0016)) at the Chirundu site, while lower coronavirus prevalence was associated with the pregnancy period (OR = 0.165 (0.044–0.567, *p* = 0.0018)).

The three viral groups analyzed at Chirundu site, ASVG-01, ASVG-05 and BSVG-01, all showed varying prevalence of CoVs, with peaks in detection during different seasons (Appendix A). For ASVG-01, the highest prevalence was detected in the 4–6 month old juvenile periods, whereas the highest peaks for ASVG-05 were during the lactation and non-gestation periods (Appendix A). For the betacoronavirus sub-viral group BSVG-01, the highest prevalence corresponded to the weaning period (Appendix A).

### 3.4. Genetic Diversity of Coronaviruses at Chirundu and Magweto Sites

A high genetic diversity of alpha and beta coronaviruses was observed in the insectivorous bat ecosystem at both sites. We amplified and sequenced a total of 532 *Coronaviruses* from both sites: 307 alphacoronaviruses (Magweto cave N = 103; Chirundu farm N = 204) and 225 betacoronaviruses (Magweto cave N = 101; Chirundu farm N = 124) (Table 2). A total of 279 out of the 532 CoV sequences were from *Hipposideros* species, 64 from *Macronycteris* spp., 89 from *Rhinolophus* spp., 17 from *Nycteris* spp. and 6 from *Miniopterus* spp. For the remaining 77 CoV sequences, the bat genus/species could not be determined mainly owing to bad sequence qualities (Table 2).

We observed different sub-viral groups with phylogenetic analyses (Figure 3 and Figure 4) which we named from A-SVG-01 to A-SVG-08 for alphacoronaviruses and B-SVG-01 to B-SVG-08 for betacoronaviruses (Table 2, Figure 3 and Figure 4).

### 3.5. Alphacoronaviruses

A-SVG-01 to A-SVG-03 groups belonged to the *Duvinacovirus* sub-genus together with the Human CoV 229E strain (Figure 3). Of the sequences comprising the A-SVG-01 group, 8 of 114 were identified at Magweto cave with four from *H. caffer* and four from *M. gigas*. The 99 remaining sequences were obtained from Chirundu farm with most of them amplified from *H. caffer* (N = 89) and the rest from *Rhinolophus* (N = 3) and *Nycteris* (N = 7) species (Table 2). Viruses belonging to the A-SVG-01 group were found at Chirundu site from August 2020 to May 2021 and at Magweto site, from March to June 2021 only (Appendix A). A-SVG-02 group was mainly composed of sequences identified in *M. gigas* (N = 41) sampled at Magweto. We also identified four other sequences obtained from Chirundu farm: two from *Rhinolophus* species and two from unclassified bat species (Figure 3, Table 2). These A-SVG-02 group sequences were collected in September and November 2020 at Magweto cave and from September to October 2020 at Chirundu site (Appendix A). The A-SVG-03 group was a subclade specific to *Hipposideros* spp. from Magweto site collected between September 2020 and March 2021. However, in this subgroup, two *Rhinolophus* and an unspecified bat species were also infected by this virus (Figure 3 and Appendix A, Table 2 and Appendix A).

The A-SVG-04 subgroup belongs to the *Setravirus* sub-genus together with the human CoV NL63 strain. Only one sequence, identified at Chirundu from a *Miniopterus mossambicus* bat species in May 2021 comprised this group (Figure 3 and Appendix A, Table 2). 

A-SVG-05 and -06 subgroups belonged to the *Decacovirus* clade. The A-SVG-05 subgroup mainly comprised sequences isolated from *Rhinolophus* bats (N = 52) from September 2020 to July 2021 at Chirundu site. Furthermore, *Hipposideros* (N = 1)*, Nycteris* (N = 1) and *Miniopterus* (N = 2) bat species were also identified as carriers of this virus at this site in the same period. At Magweto site, six A-SVG-05 sequences from *Hipposideros* species (N = 2) were identified in June 2021 and from *Macronycteris* spp. (N = 4) in April and June 2021. A-SVG-06 group sequences (N = 13) were only detected at Magweto site from *Rhinolophus* (N = 9), *Hipposideros* (N = 1), *Miniopterus* (N = 1) and from unspecified (N = 2) bat species. This viral group was present from September to November 2020 at this site (Figure 3 and Appendix A, Table 2). 

A-SVG-07 subgroup belonged to the *Minunacovirus* clade, with one sequence obtained from *Miniopterus* spp. and one from an unspecified species (Figure 3, Table 2). 

A-SVG-08 subgroup belonged to the *Rhinacovirus* clade and comprised sequences (N = 16) obtained from samples collected in August, November and December 2020 at Magweto site and in July 2021 at Chirundu site. Eight were from *Rhinolophus*, one from *Nycteris* and seven from uncharacterized bat species (Figure 3 and Appendix A, Table 2).

### 3.6. Betacoronaviruses

B-SVG-01 to B-SVG-05 groups belonged to the *Hibecovirus* clade. B-SVG-01 was the most important subclade with 105 sequences: five were from *Hipposideros* (N = 3) and *Macronycteris* (N = 2) bat species sampled at Magweto cave in March and April 2021. For the 100 remaining sequences, 73 were identified from *Hipposideros,* four from *Rhinolophus,* one from *Nycteris* and 22 from uncharacterized bat species collected from November 2020 to July 2021 (Figure 4 and Appendix A, Table 2). B-SVG-02 and -04 were specific to *Hipposideros caffer* and *Macronycteris gigas* bat species from Magweto cave and were detected from November 2020 to April 2021 and from September to June 2021, respectively (Figure 4 and Appendix A, Table 2). B-SVG-03 subclade comprised six sequences isolated at Chirundu site from *Hipposideros* (N = 2), *Rhinolophus* (N = 1) and from undetermined bat species (N = 3). This subclade was detected in February, May and June 2021 (Figure 4 and Appendix A, Table 2). B-SVG-05 was composed of 52 sequences; 51 were detected at Magweto site and one at Chirundu site. Only *Hipposideros* bat species were infected by this subclade, and these viruses were present at Magweto site from November 2020 to July 2021 and only in March 2021 at Chirundu site (Figure 4 and Appendix A, Table 2). 

B-SVG-06 and -07 belong to the *Sabercovirus* clade together with SARS-CoV-1 and -2 as well as numerous SARS-related bat CoVs isolated from *Rhinolophus* bat species in Asia and Africa. The B-SVG-06 subclade was specific to Magweto site with five sequences characterized from *Rhinolophus* bat species, one from *Nycteris* bat species and two from uncharacterized bat species, and the presence of this subclade was detected from October 2020 to March 2021 (Figure 4 and Appendix A, Table 2). As for B-SVG-07 subclade, it was specific to Chirundu site and comprised only four sequences, (N = 2) from *Rhinolophus* bat species and (N = 2) from uncharacterized bat species collected in July 2021 (Figure 4 and Appendix A, Table 2).

B-SVG-08 belonged to the Merbecovirus clade. This subclade was mainly composed of sequences isolated at Chirundu site from *Hipposideros* (N = 8), *Nycteris* (N = 6), *Rhinolophus* (N = 1) and two unspecified bat species sampled from August 2020 to March 2021 (Figure 4 and Appendix A, Table 2). Three sequences were isolated at Magweto site from *Hipposideros* (N = 2) and *Rhinolophus* (N = 1) bat species collected in April 2021.

## 4. Discussion

The majority of bat coronavirus studies in Africa mainly focused on genetic diversity and prevalence of CoVs, mostly based on single sampling sessions [43]. However, longitudinal studies covering bat reproduction cycles including the gestation, parturition, lactation, weaning and juvenile periods are crucial to further understand whether viral diversity and viral shedding vary over time and to assess the high-risk season for CoV spillovers from bats to human and domestic animal populations in close contact with these bat populations [29,43,44]. Owing to logistical challenges, few longitudinal studies have been conducted on bat coronaviruses. Moreover, these studies highlighted the variability of viral shedding over time and particularly during the reproductive season [29,31,43,45,46]. We performed a longitudinal study in two multi-bat species roosting sites in Zimbabwe. We collected fecal samples monthly from August 2020 to July 2021. Unfortunately, due to the travel restrictions of the lockdown periods during the COVID-19 pandemic, only nine months at Chirundu farm site and six months at Magweto cave site were sampled instead of the expected 12 month follow up. Nonetheless, we collected more than 3000 fecal samples covering reproductive and non-reproductive periods of the bat colonies. 

A high genetic diversity was observed among the newly identified coronaviruses including alphacoronaviruses from five sub-genera, *Duvinacovirus*, *Setracovirus, Decacovirus, Minunacovirus* and *Rhinacovirus,* as well as betacoronaviruses from three sub-genera, *Hibecovirus*, *Sarbecovirus* and *Merbecovirus*. All these sub-genera except for *Decacovirus*, *Minucavirus* and *Rhinacovirus* have been previously described in Africa [43]. Our study enlarges the knowledge on CoV diversity in bats from Africa.

The *Hipposideros* bat species were the most prominent carriers of CoVs, representing 52.4% of our CoV sequences, followed by *Rhinolophus* bats (16.7%), *Macronycteris gigas* species (12%) and more anecdotally some *Nycteris* species (3.1%) and very few *Miniopterus* bats (1.1%). Bat species were only characterized for the positive CoV samples, and it is therefore difficult to link these results with bat species community constitution at both sites during the time of this study. Nonetheless, at Chirundu farm we visually observed a shift of bat populations in June and July. Indeed, *Hipposideros* bat species were predominant before June, whereas *Rhinolophus* species became predominant in June and July, a shift confirmed by cytochrome B analyses of all coronavirus positive samples. Thus, in June and July, most of our CoV positive samples were detected from *Rhinolophus bat* spp. 

The *Duvinacovirus* sub-genus comprises the Hu-CoV 229E strain known to induce common cold in humans [32]. This virus originates from bats and more specifically from *Hipposideros* species [3]. In our study we confirmed *Hipposideros* species as the main carriers of CoV-229E-related strains in Zimbabwe. However, we also observed a *M. gigas* specific *Duvinacovirus*-related subclade. This viral strain was also detected in a few *Rhinolophus* and *Nycteris* bats, and presence in these species can be attributed to cross-species transmission from *Hipposideros* or *Macronycteris* bat species. However, cross-contamination among fecal samples collected on the floor cannot be excluded. Our phylogenetic analyses also showed phylogeographic subclades according to the sampling site.

We also identified a CoV clade belonging to the *Setracovirus* sub-genus together with the Hu-CoV NL63 strain which also induces common cold in humans [32]. To date NL63-related bat CoV have been identified to originate from the *Triaenop* bat genus [43]. In our study we characterized a NL63-CoV-related strain from a *Miniopterus* bat species. However, only one sequence was recovered, suggesting a very rare circulation of this virus in our study sites. Moreover, it is possible that this virus was acquired by this bat through contact with a *Triaenops* bat species from another site. Nonetheless, only one *Triaenops* bat species has been recorded in this region, *Triaenops afer*, which seems to be marginally present in the Eastern region of Zimbabwe [47], thus suggesting that another bat species could also be a carrier of this CoV strain in Zimbabwe. Within the *Decacovirus* sub-genus, we observed two different subclades corresponding to the two sites with the majority of *Rhinolophus* bat species as carriers of this virus. Additionally, we also observed some cross-species transmissions of this virus from *Rhinolophus* bats to *Hipposideros*, *Nycteris* and *Miniopterus* bat species, although here too cross-contamination among fecal samples collected on the floor cannot be excluded. The last two alpha subgroup viruses, A-SVG-07 and -08, were detected at Magweto and Chirundu sites, respectively, with A-SVG-07 belonging to the sub-genus *Minunacovirus* with only two sequences characterized and A-SVG-08 mainly present in *Rhinolophus* bat species belonging to the *Rhinacovirus* sub-genus. 

The *Hibecovirus* sub-genus was the most important *Betacoranavirus* clade, representing 88% of all ß-CoV sequences characterized in our study. Like the *Duvinacovirus* sub-genus, the majority of our *Hibecovirus* sequences were identified in *Hipposideros* bat species and a few *Rhinolophus* and *Nycteris* bats. The latter species might have been infected by cross-species transmission of the virus from *Hipposideros* bats. We also evidenced the presence of *Sarbecovirus* which is composed of SARS-CoV-01 and -02 strains and bat CoVs isolated from *Rhinolophus* bat species. This sub-genus was characterized at both sites. At Magweto site only eight samples were positive along with four from Chirundu, highlighting the low circulation of these viruses at both sites despite detection from October 2020 to March 2021 at Magweto site. Finally, our last ß-CoV sub viral group belonged within the *Merbecovirus* sub-genus which also comprised the MERS-CoV virus. Both *Hipposideros* and *Nycteris* bat species were infected as well as one *Rhinolophus* species. Bat CoVs from this sub-genus were mainly associated with the Vespertilioniforme bat sub-order [43,48] from *Molosidae*, *Nycteridae* and *Emballonuridae* bat families [49,50,51]. In this study we also evidenced *Merbecovirus* in the Pteropodiforme bat sub-order and particularly in *Hipposideridae* and *Rhinolophidae* bat families which extended the range of bat families as carriers of *Merbecoviruses*. 

Globally, we identified bat CoVs belonging to five out of the eight CoV sub-genera previously described for *Alphacoronavirus* genus and three out of the four sub-genera described for *Betacoronavirus* genus. We also observed what might indicate cross-species transmissions within the same site, but possibility of contamination cannot be excluded. Thus, the impact of these potential transmissions to new bat hosts has yet to be elucidated as well as whether these new hosts play a role in the dissemination of these viruses to other animal species or to humans. Additionally, we observed identical viruses in both sites suggesting that the two studied bat communities are somehow interconnected. Magweto and Chirundu sites are situated at 250 km apart, and numerous caves are present in-between, allowing bat population exchanges [52] alongside with their microorganisms over a large area. 

CoV RNA was detected in 24% of the bat samples at Chirundu site and in 16.5% of the insectivorous bat samples and 39% of the *M. gigas* bat samples at Magweto site. However, at Magweto site this result was probably underestimated since we were not able to collect samples from December 2020 to March 2021, which corresponds to the lactation and weaning periods with numerous juveniles in the cave. In other longitudinal surveys, high RNA CoV detection was found to be associated with juvenile bats [29,31,46]. Pathogens respond to the influx of susceptible hosts (new recruits) through increased transmission following the birth pulse [53,54,55]. At Chirundu site we observed a peak of RNA CoV prevalence during the lactation, weaning and juvenile periods with 37%, 44% and 35% of RNA CoV detection, respectively. We observed the same trend at Magweto site in the bats with an RNA CoV detection of 35.9% during the parturition period in November 2020 and of 34.7% during the juvenile period in March 2021. The transmission and prevalence of pathogens is facilitated by higher proportion of juveniles who are immunologically immature [31,55] explaining thus the high prevalence of CoVs during these seasons. Furthermore, the waning protection of maternal antibodies upon birth and weaning leads to increased prevalence in virus detection in bats [21]. Due to the similarity in dominant bat species between the two sites, it is likely that the Magweto site showed a similar trend in prevalence of CoVs during the unsampled and unobserved months corresponding to the lactation and weaning periods as observed at Chirundu site. Additionally, within the *M. gigas* bat species the peak of RNA CoV detection (56.4%) was in October 2020 during the gestation period which was also the period with the most important *M. gigas* population in the cave. It is important to bear in mind that this study was conducted over 1 year only, and that sampling over a long period of time would be needed to confirm the annual seasonal trends observed in this study. Moreover, the prevalence peaks of the analyzed viral subgroups which occur at different stages of the reproductive season show that the global seasonal trend observed at the CoV family level is not necessarily the case for the very viral subgroup. Further research should thus be performed on specific CoV groups which are of great importance to domestic animal and public health.

Coronavirus evolution and emergence in new hosts is driven by different factors such as recombination, horizontal gene transfer, gene duplication and alternative open reading frames which expand their functional and adaptative capacity for the current and new host. During the high viral shedding season, bats are carriers of quasi-species pools of viruses which contribute to their genetic diversity and increase their potential to jump and emerge in a new species [56]. Due to the great genetic diversity observed at both sites as well as the important RNA bat CoV prevalence during the reproductive season, our sites could be potential hotspots of new CoV spillover and potential zoonotic emergence. 

The risk of CoV spillover to human population is higher at Magweto site than Chirundu site. At Chirundu, the roosting area is in the basement of an old, damaged building with very few direct interactions between the farm workers and the bat populations. Contrary to this, Magweto site could be more susceptible to a viral spillover event. Human populations use the cave for different activities. Throughout the year, bat guano is collected inside the cave and applied as fertilizers in neighboring fields and sold for commercial earnings, generating fecal dust which is inhaled by workers. The workers and buyers are thus directly exposed to bat dejections and aerosols. Moreover, this cave is considered to be sacred, with traditional and religious activities inside the cave, and finally some people hunt *M. gigas* bats for consumption. All these cause direct interactions between bats and humans. The interface between bats and human populations at Magweto cave is thus very tight, and increased surveillance of viral circulation and zoonotic transmissions should be recommended, especially during periods with peaks of viral shedding, between October and March. 

## 5. Conclusions

Altogether, our biological results associated with the human activities at Magweto site and to a lesser extent at Chirundu site highlighted the need to further document the viral infectious risk in both human and domestic animal populations surrounding bat habitats in Zimbabwe. Particularly, the human population using caves is constantly exposed to viral shedding, and all factors required for viral spillover are met at least for the first and second stages of viral disease emergence [57,58]. This study falls within the One Health approach where the established results can be used to form a framework for further studies in human populations living in proximity to the study areas to further extrapolate the extent, if any, of the bat coronavirus circulation in humans and domestic animals through a serological study.

Our study focused on coronaviruses; however, bats host a plethora of potential infectious viruses and numerous bat species known to harbor highly fatal viruses are present in Zimbabwe. Therefore, further studies are needed to better understand the bat viral ecology in Zimbabwe and to better assess the risk of viral disease emergence in this geographic area [43,48].

## Figures and Tables

**Figure 1 viruses-14-00781-f001:**
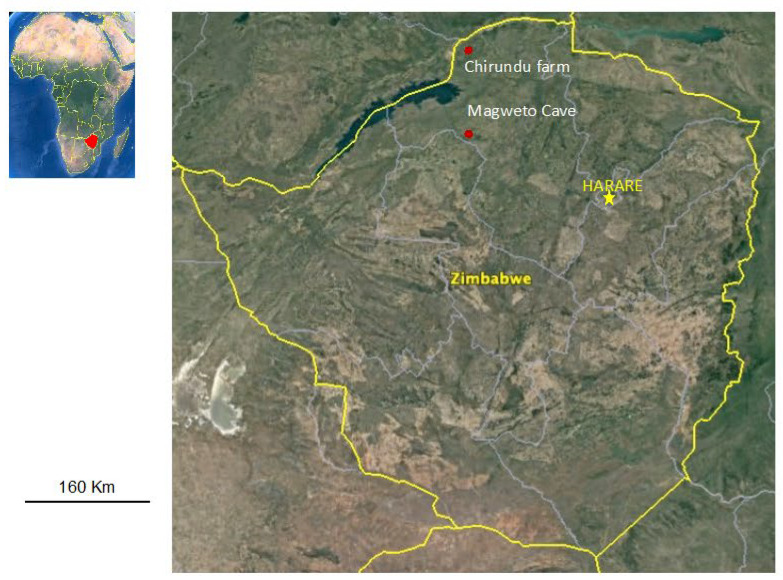
The study sites highlighted on the map show the sampling areas for insectivorous bat colonies in Hurungwe district.

**Figure 2 viruses-14-00781-f002:**
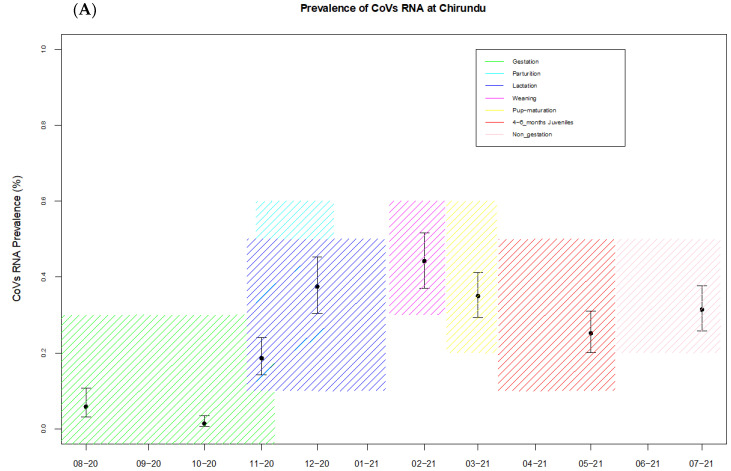
Results of the longitudinal sampling at two sites for coronaviruses. (**A**) Estimation of the coronavirus prevalence (with 95% CI) at Chirundu site, (**B**,**C**) the same for bats and *Macronycteris gigas* (respectively) at Magweto cave site. The areas colored green, cyan, navy blue, magenta, red and pink show periods of gestation, parturition, lactation, weaning, 4–6 months old juvenile and non-gestation periods, respectively, observed in the dominant bat species at both sites.

**Figure 3 viruses-14-00781-f003:**
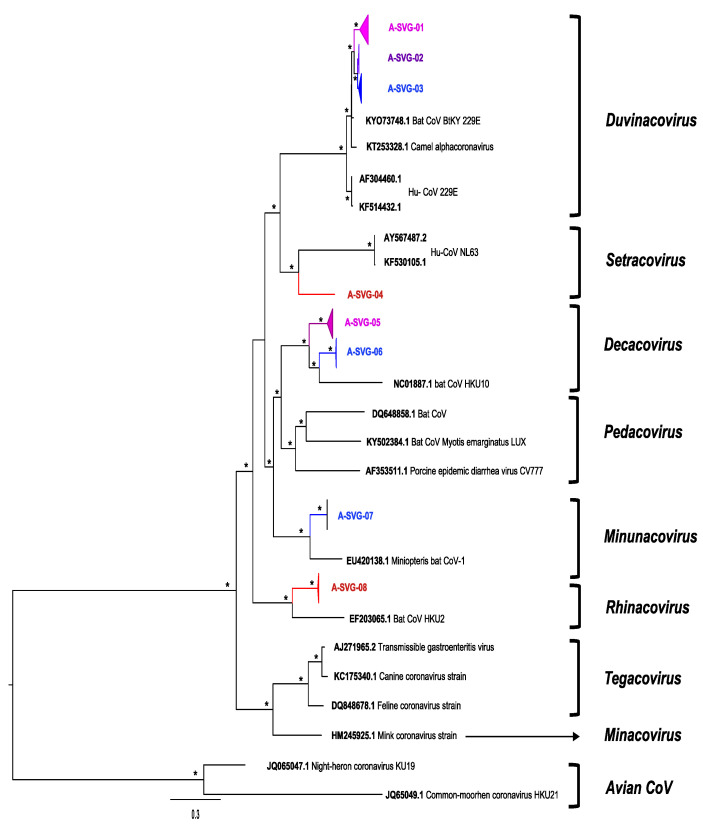
Phylogenetic tree of *AlphaCoV* partial *RdRp* gene. The sequences detected at Chirundu site are in red color and from Magweto site in blue color. Pink = more sequences detected at Chirundu than Magweto within the same subclade and purple = more sequences detected at Magweto site than Chirundu site within the subclade. The tree was built using the maximum likelihood method based on the GTR + G4 + I model. The robustness of nodes was assessed with 1000 bootstrap replicates. Bootstrap values >70 are denoted with asterisks, and those <70 are not shown.

**Figure 4 viruses-14-00781-f004:**
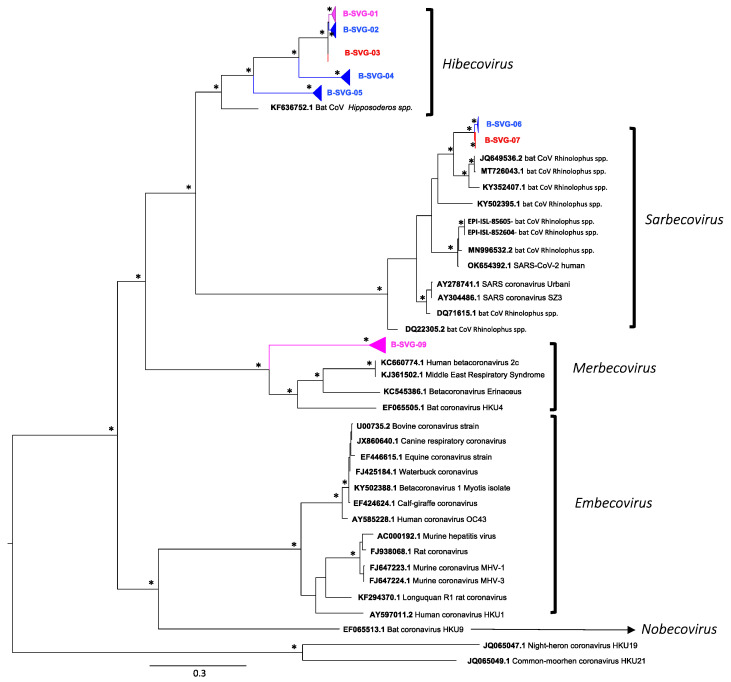
Phylogenetic tree of *BetaCoV* partial *RdRp* gene. The sequences detected at Chirundu site are in red color and at Magweto site in blue color. Pink = more sequences detected at Chirundu than Magweto site within the same subclade. The tree was built using the maximum likelihood method based on the GTR + G4 + I model. The robustness of nodes was assessed with 1000 bootstrap replicates. Bootstrap values >70 are denoted with asterisks, and those <70 are not shown.

**Table 1 viruses-14-00781-t001:** Prevalence of CoVs and confidence intervals (CIs) per month at both sites in fecal samples from insectivorous bat communities.

Site	Reproduction Cycle	Month Sampled	No of Samples Tested	No of Coronavirus Positives	Prevalence (%) + CI (95%)
Chirundu	Non-gestation	August 2020	154	9	5.8 (3.1–10.7)
	Pregnancy	October 2020	296	4	1.35 (0.5–3.4)
	Parturition	November 2020	242	45	18 (14.2–23.9)
	Lactation	December 2020	160	60	37 (30.4–45.2)
	Weaning	Februar 2021	172	76	44.2 (37–51.7)
	4–6 months juveniles	March 2021	240	84	35 (29.2–41.2)
	4–6 months juveniles	May 2021	242	61	25.2 (20.2–31)
	Non-gestation	July 2021	226	71	31 (25.7–37.7)
Overall prevalence			1732	410	23.7 (21.73–25.73)
Magweto	Non-gestation	September 2020	348	7	2.0 (0.98–4.1)
	Pregnancy	October 2020	257	27	10.5 (7.3–14.9)
	Parturition	November 2020	228	82	35.9 (30–42.4)
	4–6 months juveniles	March 2021	242	84	34.7 (28.9–40.9)
	4–6 months juveniles	April 2021	241	40	16.6 (12.4–21.8)
	Non-gestation	June 2021	242	17	7.02 (4.4–10.9)
Overall prevalence			1558	257	16.5 (14.74–18.42)
Magweto *M. gigas*	Non-gestation	September 2020	2	2	100 (34.2–100)
	Pregnancy	Octobe 2020	124	70	56.4 (47.7–64.9)
	Parturition	November 2020	74	26	35.1 (25.2–46.5)
	4–6 months juveniles	March 2021	42	8	19.1 (9.98–33.3)
	4–6 months juveniles	April 2021	39	9	23.1 (12.7–38.3)
	Non-gestation	June 2021	27	5	18.5 (8.18–36.7)
Overall prevalence			308	120	38.96 (33.68–44.51)

**Table 2 viruses-14-00781-t002:** Longitudinal detection of different viral groups in Magweto and Chirundu sites.

		Number of Sequence per Site		Number of Sequences per Bat Species						Longitudinal Detection of the Different Viral Groups	
Viral Group	Total Number of Sequences	Magweto	Chirundu	Hipposideros spp.	Macronycteris spp.	Rhinolophe spp.	Nycteris spp.	Miniopterus spp.	Unknown	Magweto	Chirundu
A-SVG-01	114	8	106	89	4	3	7	0	11	March 2021 to June 2021	August 2020 to May 2021
A-SVG-02	45	42	3	0	41	2	0	0	2	September 2020 to November 2020	October 2020 to November 2020
A-SVG-03	32	32	0	29	0	2	0	0	1	September 2020 to March 2021	-
A-SVG-04	1	0	1	0	0	0	0	1	0	-	May 2021
A-SVG-05	84	6	78	7	4	52	1	3	17	June 2021	September 2020 to July 2021
A-SVG-06	13	13	0	1	0	9	0	1	2	September 2020 to November 2020	-
A-SVG-07	2	2	0	0	0	0	0	1	1	October 2020	-
A-SVG-08	16	0	16	0	0	8	1	0	7	-	August-November-December 2020-July 2021
**sub Total**	**307**	**103**	**204**	**126**	**49**	**76**	**9**	**6**	**41**		
		**Number of Sequence per site**		**Number of sequences per Bat species**						**Longitudinal Detection of the different viral groups**	
B-SVG-01	105	5	100	76	2	4	1	0	22	March to April 2021	November 2020 to July 2021
B-SVG-02	21	21	0	19	0	0	0	0	2	November 2020 to April 2021	-
B-SVG-03	6	0	6	2	0	1	0	0	3	-	February 2021 May and June 2021
B-SVG-04	13	13	0	0	13	0	0	0	0	September 2020 to June 2021	-
B-SVG-05	51	51	0	48	0	0	0	0	3	November 2020 to July 2021	March 2021
B-SVG-06	8	8	0	0	0	5	1	0	2	October 2020 to March 2021	-
B-SVG-07	4	0	4	0	0	2	0	0	2	-	July 2021
B-SVG-08	17	3	14	8	0	1	6	0	2	April 2021	August 2020 to Mach 2021
**sub Total**	**225**	**101**	**124**	**153**	**15**	**13**	**8**	**0**	**36**		
**TOTAL**	**532**	**204**	**328**	**279**	**64**	**89**	**17**	**6**	**77**		

## Data Availability

Not applicable.

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
