# Peer review of "Longitudinal Survey of Coronavirus Circulation and Diversity in Insectivorous Bat Colonies in Zimbabwe"

_viruses, 2022, doi:10.3390/v14040781_

Round 1
Reviewer 1 Report
Longitudinal survey of Coronavirus circulation and diversity in insectivorous bat colonies in Zimbabwe
A relatively unique, longitudinal study design, including the gene-based species identification of the bat. A large number of fecal samples collected.
This study is deserving of publication in Viruses. A few questions/comments:
How sensitive do the investigators believe is the identification of fecal CV sequences reflect bat infection? Might some bats infected with CV not shed viral RNA in the feces? Might some CV vRNA degrade within the feces? Are all bat CVs shed in feces?
Do the investigators believe that these CVs are pathogenic or apathogenic in bats? What systems do they believe are infected (GI alone) ? Do the investigators believe that bat feces contain infectious virions?
Line 128-129
Justification for this assumption?
Lines 170-171
Target gene/genetic locus for PCR? Volume of first round rxn carried over to second? Positive and negative control reactions run in parallel?
Line 240-242
Different font/font size
Line 244-247
Punctuation
Lines 250-252
Provide years
Figure 2
It is difficult to read / interpret these figures (especially the inset). Enlarge (one figure/line)?
Author Response
We thank the reviewer for these accurate comments and questions.
Author’ responses:
How sensitive do the investigators believe is the identification of fecal CV sequences reflect bat infection?
Response: The objective of our study was to determine the prevalence and dynamics of CoV infections over time. Our results clearly indicate that the prevalence of CoV infections changes over time and is correlated with the reproductive phenology of these animals. The PCR system implemented in this study has largely demonstrated its effectiveness for the detection of different Alpha and Beta coronavirus sub-genera. However, the effectiveness of gene amplification depends on different factors and in particular to the detection limit (LOD) of the system. In fact, if few virus particles are present in the sample, it will be difficult to detect them with a conventional PCR system. However, we believe that our study is representative of viral dynamics in the 2 sites studied.
Might some bats infected with CV not shed viral RNA in the feces? What systems do they believe are infected (GI alone)? Are all bat CVs shed in feces?
Response: Bat coronaviruses are predominantly excreted in faecal material, which creates opportunities for exposure to other individuals and species, and facilitates easy transmission throughout bat populations. Bat coronaviruses have also been reported at low frequencies in oral swabs, urine (Mendenhall et al., 2017) and tissues such as rectum and intestine (Geldenhuys et al, 2018), lung (Shehata et al., 2016), spleen and brain (Anindita et al., 2015).
Might some CV vRNA degrade within the feces?
Response:Yes obviously. Coronaviruses are RNA enveloped virus and are less resistant than non-enveloped virus. This is why we collected very fresh feces.
Do the investigators believe that these CVs are pathogenic or apathogenic in bats?
Response: According to the literature, bats are the natural reservoir host of Coronavirus and reservoir hosts of zoonoses appear tolerant of the pathogenic effects of infection. Whether bat species are universally tolerant of coronavirus infection remains unclear as few experimental coronavirus challenge studies involving bats have been performed, the putative natural reservoir bat species was often not used and it is unclear whether the infectious doses resembled those of natural exposures. In bats experimentally infected with coronaviruses, some individuals have shown mild tissue damage, including rhinitis and interstitial pneumonia, with virus or viral RNA detected in the respiratory tract and/or intestines; however, infected animals did not exhibit evident clinical signs of infection.
Do the investigators believe that bat feces contain infectious virions?
Response: Yes, different studies successfully isolated bat Coronavirus from faecal samples (e.g., Xing-Yi Ge et al, Nature 2013)
Line 128-129: Justification for this assumption?
Response: Due to the very large number of bats in the two study sites we consider it very unlikely that we collected the same individual using this sampling protocol, but it remains a possibility. The only way to ensure that each stool corresponds to an individual would be either to capture the animals with a tagging system or to do a microsatellite analysis which is very time consuming and expensive.
Lines 170-171:
Target gene/genetic locus for PCR? :
Response: Line 164: 2.7. RNA-dependent RNA-polymerase (RdRp) nested-PCR on Coronaviruses using Pan-coronavirus primers
Line 166: CoVs partial RdPd gene were amplified by nested PCR…..
Volume of first round rxn carried over to second?
Response: Added line 166-167
Positive and negative control reactions run in parallel?
Response:Added line 182-183
Line 240-242: Different font/font size
Response:Corrected
Line 244-247: Punctuation
Response:Corrected
Lines 250-252: Provide years
Response:Added in the manuscript
Figure 2
It is difficult to read / interpret these figures (especially the inset). Enlarge (one figure/line)?
Response: Corrected
Reviewer 2 Report
It is an interesting work in a country where there are not a lot of data about CoV. The virological results are very interesting. The work planning is good because in order to understand the CoV dynamics longitudinal studies are needed. However, the work has some limitations as a consequence of the methodology used. In multi-specific colonies the collection of feces on the ground has limitations. It is not clear whether there were any other species in the colonies other than those mentioned. I miss the description of the dynamics of the colonies in terms of number of individuals. This it is very important in virus dynamics.
On the other hand, I miss the species determination for the unknown bat group. It is a pity that the species of the unknown bat group have not been determined since in this bat pool you have found 41 AlphaCoV and 36 betaCoV.
Introduction
Page 2, line 52. Please, change “six coronaviruses” by “seven coronaviruses”
You can complete the introduction with the reference:
Ar Gouilh M, Puechmaille SJ, Diancourt L, Vandenbogaert M, Serra-Cobo J, Lopez Roïg M, Brownh P, Moutou F, Caroa V, Vabret A, Manuguerra JC. 2018. SARS-CoV related Betacoronavirus and diverse Alphacoronavirus members found in western old-world. Virology, 517: 88–97 https://doi.org/10.1016/j.virol.2018.01.014
Circulation of viruses in bat populations is also affected by number of species in the colony and increase with the presence of migratory species (Serra-Cobo J, López-Roig M. 2016. Bats and emerging infections: an ecological and virological puzzle. Diseases and Public Health, 972: 35-48. DOI: 10.1007/5584_2016_131).
Material and methods
Lines 140-141. When have been done these two capture sessions?
Results
To talk about small and large bats is imprecise. I think that it would be better to write M. gigas colony for large insectivorous bats and, if possible, to look for another term for the small insectivorous bats.
Line 289. Why do you write Hipposideros spp., Rhinolophus spp., Nycteris spp. and Miniopterus spp. without to specify the species? It's the same question for the table 2.
Please add the scale to Figure 3.
Line 427. Please delete the sentence:
“For this study …….. defaecation area.”
It has already been explained in material and methods, it is repetitive.
Lines 512-514. The increase in the prevalence can be consequence not only the immature of immunity system of juveniles but also to incorporate the juveniles in the colony. The number of susceptible increase and next also the prevalence.
Author Response
We thank the reviewer for these accurate comments and questions.
Comments and Suggestions for Authors
It is an interesting work in a country where there are not a lot of data about CoV. The virological results are very interesting. The work planning is good because in order to understand the CoV dynamics longitudinal studies are needed. However, the work has some limitations as a consequence of the methodology used. In multi-specific colonies the collection of feces on the ground has limitations. It is not clear whether there were any other species in the colonies other than those mentioned. I miss the description of the dynamics of the colonies in terms of number of individuals. This it is very important in virus dynamics. On the other hand, I miss the species determination for the unknown bat group. It is a pity that the species of the unknown bat group have not been determined since in this bat pool you have found 41 AlphaCoV and 36 betaCoV.
Response:
Thank you for your insightful comments. As you point out the non-invasive approach has its limits. Indeed, we cannot exclude the presence of other bat species which may be in low numbers or simply CoV negative. In fact, we only identified the positive CoV bat species. However, we have successfully sequenced the CYTB and/or 12S RNA genes on 463 samples which represents 13% of our overall sampling.
We did not assess the number of individuals. This should have been done at different periods because we observed changes in the species present on the sites.
For the unidentified species, we used both CYTB and 12S RNA systems and we modified the PCR parameters to try to obtain them but without success.
Introduction
Page 2, line 52. Please, change “six coronaviruses” by “seven coronaviruses”
Response: Corrected.
You can complete the introduction with the reference:
Ar Gouilh M, Puechmaille SJ, Diancourt L, Vandenbogaert M, Serra-Cobo J, Lopez Roïg M, Brownh P, Moutou F, Caroa V, Vabret A, Manuguerra JC. 2018. SARS-CoV related Betacoronavirus and diverse Alphacoronavirus members found in western old-world. Virology, 517: 88–97 https://doi.org/10.1016/j.virol.2018.01.014
Circulation of viruses in bat populations is also affected by number of species in the colony and increase with the presence of migratory species (Serra-Cobo J, López-Roig M. 2016. Bats and emerging infections: an ecological and virological puzzle. Diseases and Public Health, 972: 35-48. DOI: 10.1007/5584_2016_131).
Response: We Added the first one.
Material and methods
Lines 140-141. When have been done these two capture sessions?
Response: Information added in the manuscript line 141-142
Results
To talk about small and large bats is imprecise. I think that it would be better to write M. gigas colony for large insectivorous bats and, if possible, to look for another term for the small insectivorous bats.
Response: We corrected in the manuscript
Line 289. Why do you write Hipposideros spp., Rhinolophus spp., Nycteris spp. and Miniopterus spp. without to specify the species? It's the same question for the table 2.
Response: We used 2 different PCR systems to determine the bat species. Nonetheless by using CYT B system we were able to identify the species (see Fig B1) whereas by using the 12S RNA system we only characterize the genus. We decided to use the genus name instead of the species name because both 12S RNA and CYT B results are used in the study.
Please add the scale to Figure 3
Response: We added the scale
Line 427. Please delete the sentence:
“For this study …….. defaecation area.”
It has already been explained in material and methods, it is repetitive.
Response: Sentence removed
Lines 512-514. The increase in the prevalence can be consequence not only the immature of immunity system of juveniles but also to incorporate the juveniles in the colony. The number of susceptible increase and next also the prevalence.
Response: We agree with this comment: Line 525 “Pathogens respond to the influx of susceptible hosts (new recruits) through increased transmission following the birth pulse”
Reviewer 3 Report
Chidoti performed longitudinal survey of CoVs circulating in two colonies in Zimbabwe. The study provides novel insights for bats harboring diversified CoVs, expanding to more bat host species. The study could be potentially improved.
1) The authors should provide NGS for the novel CoVs detected in these bat samples or at least to show several representative novel CoV genome structures, i.e. the novel ones in phylogenies. More importantly, the complete genomes would reveal so much info.
2) The phylogenetic analyses could be improved by adding more reference sequences.
3) The images should be improved as they look a little simpler and less informative in their current forms.
4) The two locations are not far away parted. Are there any viral cross-species transmission events occurred among them?
Author Response
Comments and Suggestions for Authors
Chidoti performed longitudinal survey of CoVs circulating in two colonies in Zimbabwe. The study provides novel insights for bats harboring diversified CoVs, expanding to more bat host species. The study could be potentially improved.
1) The authors should provide NGS for the novel CoVs detected in these bat samples or at least to show several representative novel CoV genome structures, i.e. the novel ones in phylogenies. More importantly, the complete genomes would reveal so much info.
Response:
We agree on the interest of sequencing the complete genomes of these viruses and on the information that this would bring but it is not the object of this study. Nonetheless the purpose of this work was to investigate the prevalence and circulation of coronaviruses in two different sites in Zimbabwe, to which we responded. In addition, we are working on the amplification of the complete genomes of coronaviruses (among others) using NGS methods (Illumina and MiNion) combined with classical Sanger-type methods. This is the subject of another project which will take between 1 and 2 years to complete and whose results will be submitted for publication. The implementation of this type of sequencing is not so simple. For example, in a previous study, we used an unbiased NGS approach on a CoV positive bat faeces sample. Seventy percent of the viral sequences obtained were insect viruses, resulting of bat diets (Bourgarel et al., 2019), and only 3 were CoV sequences.
Therefore, in this study according to its aim we will not generate additional Coronavirus genetic data.
2) The phylogenetic analyses could be improved by adding more reference sequences.
Response:
In the first phylogenetic analyses we used a very wide range of references. We then selected the most relevant references. The addition of references would not bring any additional information and would not modify the interpretation of our results.
3) The images should be improved as they look a little simpler and less informative in their current forms.
Responses: Can you please give us more informations/details. Which images? What do you mean by simpler? We used classical R graphic for the prevalence graphs and classical phylogenetic topologies. Do you have propositions to improve these images? We enlarge the Figure 2 A,B,C in order to render it more readable.
4) The two locations are not far away parted. Are there any viral cross-species transmission events occurred among them?
Response:
We discussed this point (Line 492-502): “Globally, we identified bat CoVs belonging to five out of the eight CoV sub-genera previously described for Alphacoronavirus genus, and three out of the four sub-genera described for Betacoronavirus genus. We also observed what might indicate cross-species transmissions within the same site, but possibility of contamination cannot be excluded. Thus, the impact of these potential transmissions to new bat hosts has yet to be elucidated as well as establishing whether these new hosts might play a role in the dissemination of these viruses to other animal species or yet to humans. Additionally, we observed identical viruses in both sites suggesting that the two studied bat communities are somehow interconnected. Magweto and Chirundu sites are situated at 250 km apart and numerous caves are present in-between allowing bat population exchanges [52] alongside with their microorganism, over a large area».
Round 2
Reviewer 2 Report
Your comments clarify the read of paper. It is a good article which deserves to be published. I encourage to pursue the work in this country.
Reviewer 3 Report
The authors have addressed my concern. I have no further comments.